# Approximate Knowledge Compilation by Online Collapsed Importance Sampling

**Tal Friedman**
Computer Science Department
University of California
Los Angeles, CA 90095
tal@cs.ucla.edu

**Guy Van den Broeck**
Computer Science Department
University of California
Los Angeles, CA 90095
guyvdb@cs.ucla.edu

## Abstract

We introduce collapsed compilation, a novel approximate inference algorithm for discrete probabilistic graphical models. It is a collapsed sampling algorithm that incrementally selects which variable to sample next based on the partial compilation obtained so far. This online collapsing, together with knowledge compilation inference on the remaining variables, naturally exploits local structure and context-specific independence in the distribution. These properties are used implicitly in exact inference, but are difficult to harness for approximate inference. Moreover, by having a partially compiled circuit available during sampling, collapsed compilation has access to a highly effective proposal distribution for importance sampling. Our experimental evaluation shows that collapsed compilation performs well on standard benchmarks. In particular, when the amount of exact inference is equally limited, collapsed compilation is competitive with the state of the art, and outperforms it on several benchmarks.

## 1  Introduction

Modern probabilistic inference algorithms for discrete graphical models are designed to exploit key properties of the distribution. In addition to classical conditional independence, they exploit local structure in the individual factors, determinism coming from logical constraints (Darwiche, 2009), and the context-specific independencies that arise in such distributions (Boutilier et al., 1996). The *knowledge compilation* approach in particular forms the basis for state-of-the-art probabilistic inference algorithms in a wide range of models, including Bayesian networks (Chavira & Darwiche, 2008), factor graphs (Choi et al., 2013), statistical relational models (Chavira et al., 2006; Van den Broeck, 2013), probabilistic programs (Fierens et al., 2015), probabilistic databases (Van den Broeck & Suciu, 2017), and dynamic Bayesian networks (Vlasselaer et al., 2016). Based on logical reasoning techniques, knowledge compilation algorithms construct an *arithmetic circuit* representation of the distribution on which inference is guaranteed to be efficient (Darwiche, 2003). The inference algorithms listed above have one common limitation: they perform exact inference by compiling a worst-case exponentially-sized arithmetic circuit representation. Our goal in this paper is to upgrade these techniques to allow for *approximate probabilistic inference*, while still naturally exploiting the structure in the distribution. We aim to open up a new direction towards scaling up knowledge compilation to larger distributions.

When knowledge compilation produces circuits that are too large, a natural solution is to sample some random variables and do exact compilation on the smaller distribution over the remaining variables. This collapsed sampling approach suffers from two problems. First, collapsed sampling assumes that one can determine a priori which variables need to be sampled to make the distribution amenable to exact inference. When dealing with large amounts of context-specific independence, it is difficult to

find such a set, because the independencies are a function of the particular values that variables get instantiated to. Second, collapsed sampling assumes that one has access to a proposal distribution that determines how to sample each variable, and the success of inference largely depends on the quality of this proposal. In practice, the user often needs to specify the proposal distribution manually, and it is difficult to automatically construct one that is general purpose.

As our first contribution, Section 2 introduces *online collapsed importance sampling*, where the sampler chooses which variable to sample next based on the values sampled for previous variables. This algorithm is a solution to the first problem identified above: based on the context of each individual sample, it allows the sampler to determine which subset of the variables is amenable to exact inference. We show that the sampler corresponds to a classical collapsed importance sampler on an augmented graphical model and prove conditions for it to be asymptotically unbiased.

Section 3 describes our second contribution: a *collapsed compilation* algorithm that maintains a partially-compiled arithmetic circuit during online collapsed importance sampling. This circuit provides a solution to the second problem identified above: it serves as a highly-effective proposal distribution at each step of the algorithm. Moreover, by setting a limit on the circuit size as we compile more factors into the model, we are able to sample exactly as many variables as needed to fit the arithmetic circuit into memory. This allows us to maximize the amount of exact inference performed by the algorithm. Crucially, through online collapsing, the set of collapsed variables changes with every sample, exploiting different independencies in each sample's arithmetic circuit. We provide an open-source Scala implementation of this collapsed compilation algorithm.[1]

Finally, we experimentally validate the performance of collapsed compilation on standard benchmarks. We begin by empirically examining properties of collapsed compilation, to show the value of the proposal distribution and pick apart where performance improvements are coming from. Then, in a setting where the amount of exact inference is fixed, we find that collapsed compilation is competitive with state-of-the-art approximate inference algorithms, outperforming them on several benchmarks.

## 2  Online Collapsed Importance Sampling

We begin with a brief review of collapsed importance sampling, before motivating the need for dynamically selecting which variables to sample. We then demonstrate that we can select variables in an online fashion while maintaining the desired unbiasedness property of the sampler, using an algorithm we call online collapsed importance sampling.

We denote random variables with uppercase letters ($X$), and their instantiation with lowercase letters ($x$). Bold letters denote sets of variables ($\mathbf{X}$) and their instantiations ($\mathbf{x}$). We refer to Koller & Friedman (2009) for notation and formulae related to (collapsed) importance sampling.

### 2.1  Collapsed Importance Sampling

The basic principle behind collapsed sampling is that we can reduce the variance of an estimator by making part of the inference exact. That is, suppose we partition our variables into two sets: $\mathbf{X_p}$, and $\mathbf{X_d}$. In collapsed importance sampling, the distribution of variables in $\mathbf{X_p}$ will be estimated via importance sampling, while those in $\mathbf{X_d}$ will be estimated by computing exactly $P(\mathbf{X_d}|\mathbf{x_p})$ for each sample $\mathbf{x_p}$. In particular, suppose we have some function $f(\mathbf{x})$ where $\mathbf{x}$ is a complete instantiation of $\mathbf{X_p} \cup \mathbf{X_d}$, and a proposal distribution $Q$ over $\mathbf{X_p}$. Then we estimate the expectation of $f$ by

$$\hat{\mathbb{E}}(f) = \frac{\sum_{m=1}^{M} w[m](\mathbb{E}_{P(\mathbf{X_d}|\mathbf{x_p}[m])}[f(\mathbf{x_p}[m], \mathbf{X_d})])}{\sum_{m=1}^{M} w[m]} \tag{1}$$

on samples $\{\mathbf{x_p}[m]\}_{m=1}^{M}$ drawn from a proposal distribution $Q$. For each sample, we analytically compute the importance weights $w[m] = \frac{\hat{P}(\mathbf{x_p}[m])}{Q(\mathbf{x_p}[m])}$, and the exact expectation of $f$ conditioned on the sample, that is, $\mathbb{E}_{P(\mathbf{X_d}|\mathbf{x_p}[m])}[f(\mathbf{x_p}[m], \mathbf{X_d})]$. Due to the properties of importance samplers, the estimator given by (1) is asymptotically unbiased. Moreover, if we compute $P(\mathbf{x_p}[m])$ exactly rather than the unnormalized $\hat{P}(\mathbf{x_p}[m])$, then the estimator is unbiased (Tokdar & Kass, 2010).

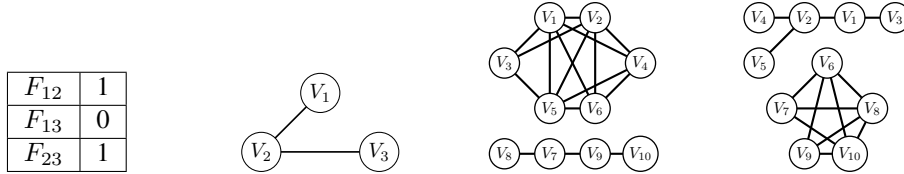

| | |
|---|---|
| $F_{12}$ | 1 |
| $F_{13}$ | 0 |
| $F_{23}$ | 1 |

(a) Sampled Friendships   (b) Induced Dependencies   (c) Induced Network $G_1$   (d) Induced Network $G_2$

Figure 1: Different samples for $\mathbf{F}$ can have a large effect on the resulting dependencies between $\mathbf{V}$.

## 2.2 Motivation

A critical decision that needs to be made when doing collapsed sampling is selecting a partition – which variables go in $\mathbf{X_p}$ and which go in $\mathbf{X_d}$. The choice of partition can have a large effect on the quality of the resulting estimator, and the process of choosing such a partition requires expert knowledge. Furthermore, selecting a partition a priori that works well is not always possible, as we will show in the following example. All of this raises the question whether it is possible to choose the partition on the fly for each sample, which we will discuss in Section 2.3.

Suppose we have a group of $n$ people, denoted $1, ..., n$. For every pair of people $(i, j), i < j$, there is a binary variable $F_{ij}$ indicating whether $i$ and $j$ are friends. Additionally, we have features $V_i$ for each person $i$, and $F_{ij} = 1$ (that is $i, j$ are friends) implies that $V_i$ and $V_j$ are correlated. Suppose we are performing collapsed sampling on the joint distribution over $\mathbf{F}$ and $\mathbf{V}$, and that we have already decided to place all friendship indicators $F_{ij}$ in $\mathbf{X_p}$ to be sampled. Next, we need to decide which variables in $\mathbf{V}$ to include in $\mathbf{X_p}$ for the remaining inference problem over $\mathbf{X_d}$ to become tractable. Observe that given a sampled $\mathbf{F}$, due to the independence properties of $\mathbf{V}$ relying on $\mathbf{F}$, a graphical model $G$ is induced over $\mathbf{V}$ (see Figures 1a,1b). Moreover, this graphical model can vary greatly between different samples of $\mathbf{F}$. For example, $G_1$ in Figure 1c densely connects $\{ V_1, ..., V_6 \}$ making it difficult to perform exact inference. Thus, we will need to sample some variables from this set. However, exact inference over $\{ V_7, ..., V_{10} \}$ is easy. Conversely, $G_2$ in Figure 1d depicts the opposite scenario: $\{ V_1, ..., V_5 \}$ forms a tree, which is easy for inference, whereas $\{ V_6, ..., V_{10} \}$ is now intractable. It is clearly impossible to choose a small subset of $\mathbf{V}$ to sample that fits all cases, thus demonstrating a need for an online variable selection during collapsed sampling.

## 2.3 Algorithm

We now introduce our online collapsed importance sampling algorithm. It decides at sampling time which variables to sample and which to do exact inference on.

To gain an intuition, suppose we are in the standard collapsed importance sampling setting. Rather than sampling an instantiation $\mathbf{x_p}$ jointly from $Q$, we can instead first sample $x_{p_1} \sim Q(X_{p_1})$, then $x_{p_2} \sim Q(X_{p_2}|x_{p_1})$, and so on using the chain rule of probability. In *online* collapsed importance sampling, rather than deciding $X_{p_1}, X_{p_2}, X_{p_3}, \ldots$ a priori, we select which variable will be $X_{p_2}$ based on the previous sampled value $x_{p_1}$, we select which will be $X_{p_3}$ based on $x_{p_1}$ and $x_{p_2}$, and so on.

**Definition 1.** *Let $\mathbf{y}$ be an instantiation of $\mathbf{Y} \subset \mathbf{X}$. A variable selection policy $\pi$ takes $\mathbf{y}$ and either stops sampling or returns a distribution over which variable in $\mathbf{X} \setminus \mathbf{Y}$ should be sampled next.*

For example, a naive policy could be to select a remaining variable uniformly at random. Once the policy $\pi$ stops sampling, we are left with an instantiation $\mathbf{x_p}$ and a set of remaining variables $\mathbf{X_d}$, where both are specific to the choices made for that particular sample.

Algorithm 1 shows more precisely how online collapsed importance sampling generates a single sample, given a full set of variables $\mathbf{X}$, a variable selection policy $\pi$, and proposal distributions $Q_{X_i|\mathbf{x_p}}$ for any

---

**Algorithm 1:** Online Collapsed IS

**Input:** $\mathbf{X}$: The set of all variables,
$\quad\quad\quad$ $\pi$: Variable selection policy,
$\quad\quad\quad$ $Q_{X_i|\mathbf{x_p}}$: Proposal distributions

**Result:** A sample $\left( \mathbf{X_d^m}, \mathbf{x_p^m}, w[m] \right)$

1 $\mathbf{x_p} \leftarrow \{\}$ ; $\mathbf{X_d} \leftarrow \mathbf{X}$
2 **while** $\pi$ *does not stop* **do**
3 $\quad$ $X_i \sim \pi\left(\mathbf{x_p}\right)$
4 $\quad$ $x_i \sim Q_{X_i|\mathbf{x_p}}(X_i|\mathbf{x_p})$
5 $\quad$ $\mathbf{x_p} \leftarrow \mathbf{x_p} \cup \{x_i\}$
6 $\quad$ $\mathbf{X_d} \leftarrow \mathbf{X_d} \setminus \{X_i\}$

7 **return** $\left( \mathbf{X_d}, \mathbf{x_p}, \frac{\hat{P}(\mathbf{x_p})}{Q(\mathbf{x_p})} \right)$

---

choice of $X_i$ and $\mathbf{x_p}$. This sample consists of a set of variables $\mathbf{X_d^m}$ to do exact inference for, an instantiation of the sampled variables $\mathbf{x_p^m}$, and the corresponding importance weights $w[m]$, all indexed by the sample number $m$. Note that $\mathbf{x_p}$ is a set of variables together with their instantiations, while $\mathbf{X_d}$ is just a set of variables. The global joint proposal $Q(\mathbf{x_p})$, denoting the probability that Algorithm 1 returns $\mathbf{x_p}$, is left abstract for now (see Section 2.4.2 for a concrete instance). In general, it is induced by variable selection policy $\pi$ and the individual local proposals $Q_{X_i|\mathbf{x_p}}$.

**Definition 2.** *Given $M$ samples $\left\{ \mathbf{X_d^m}, \mathbf{x_p^m}, w[m] \right\}_{m=1}^{M}$ produced by online collapsed importance sampling, the* online collapsed importance sampling estimator *of $f$ is*

$$\hat{\mathbb{E}}(f) = \frac{\sum_{m=1}^{M} w[m] (\mathbb{E}_{P(\mathbf{X_d^m}|\mathbf{x_p^m})}[f(\mathbf{x_p^m}, \mathbf{X_d^m})])}{\sum_{m=1}^{M} w[m]}. \tag{2}$$

Note that the only difference compared to Equation 1 is that sets $\mathbf{X_p^m}$ and $\mathbf{X_d^m}$ vary with each sample.

## 2.4 Analysis

Our algorithm for online collapsed importance sampling raises two questions: does Equation 2 yield unbiased estimates, and how does one compute the proposal $Q(\mathbf{x_p})$? We study both questions next.

### 2.4.1 Unbiasedness of Estimator

If we let $\pi$ be a policy that always returns the same variables in the same order, then we recover classical offline collapsed importance sampling - and thus retain all of its properties. In order to make a similar statement for any arbitrary policy $\pi$, we will use the augmented factor graph construction presented in Figure 2. Our goal is to reduce online collapsed importance sampling on $F$ to a problem of doing offline collapsed importance sampling on $F_A$.

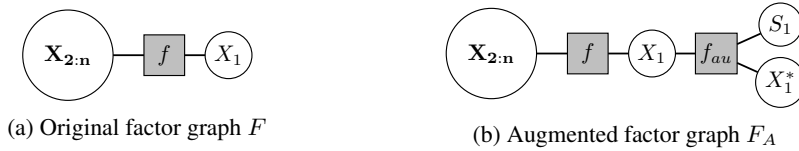

(a) Original factor graph $F$      (b) Augmented factor graph $F_A$

Figure 2: Online collapsed sampling corresponds to collapsed sampling on an augmented graph

Intuitively, we add variable $X_i^*$ to the factor graph, representing a copy variable of $X_i$. We design our offline collapsed sampler on augmented graph $F_A$ such that we are always sampling $X_i^*$ and computing $X_i$ exactly. To make this possible without actually inferring the entire distribution exactly, we add variable $S_i$ to the model (also always to be sampled). Each $S_i$ acts as an indicator for whether $X_i^*$ and $X_i$ are constrained to be equal. $S_i$ can also be thought of as indicating whether or not we are sampling $X_i$ in our original factor graph $F$ when doing online collapsed importance sampling. These dependencies are captured in the new factor $f_{au}$. We are now ready to state the following results.

**Theorem 1.** *For any factor graph $F$ and its augmented graph $F_A$, we have $\forall \mathbf{x}$ , $P_F(\mathbf{x}) = P_{F_A}(\mathbf{x})$.*

**Theorem 2.** *Let $F$ be a factor graph and let $F_A$ be its augmented factor graph. The collapsed importance sampling estimator (Eq. 1) with $\mathbf{X_p} = \mathbf{X}^* \cup \mathbf{S}$ and $\mathbf{X_d} = \mathbf{X}$ on $F_A$ is equivalent to the online collapsed importance sampling estimator (Eq. 2) on $F$.*

**Corollary 1.** *The estimator given by Eq. 2 is asymptotically unbiased.*

Proofs and the details of this construction can be found in Appendix A.

### 2.4.2 Computing the Proposal Distribution

Our next question is how to compute the global joint proposal distribution $Q(\mathbf{x_p})$, given that we have variable selection policy $\pi$ and each local proposal distribution $Q_{X_i|\mathbf{x}_p}$. Notice that since these $Q_{X_i|\mathbf{x}_p}$ are unconstrained and unrelated distributions, the computation is not easy in general. In particular, considering $|\mathbf{X_p}| = n$ and our previous example of a uniformly random policy $\pi$, then for any given instantiation $\mathbf{x_p}$, there are $n!$ different ways $\mathbf{x_p}$ could be sampled by Algorithm 1 – one for

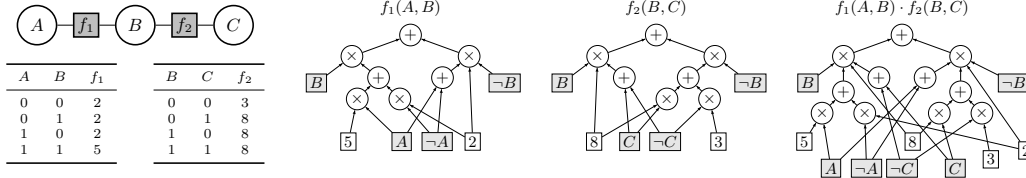

Figure 3: Multiplying Arithmetic Circuits: Factor graph and ACs for individual factors which multiply into a single AC for the joint distribution. Given an AC, inference is tractable by propagating inputs.

each ordering that arrives at $\mathbf{x_p}$. In this case, computing $Q(\mathbf{x_p})$ requires summing over exponentially many terms, which is undesirable. Instead, we restrict the variable selection policies we use to the following class.

**Definition 3.** *A* deterministic *variable selection policy* $\pi(\mathbf{x_p})$ *is a function with a range of* $\mathbf{X} \setminus \mathbf{X_p}$.

**Theorem 3.** *For any sample* $\mathbf{x_p}$ *and deterministic variable selection policy* $\pi(\mathbf{x_p})$, *there is exactly one order* $X_{p_1}, X_{p_2}, \ldots, X_{p_{|\mathbf{x_p}|}}$ *in which the variables* $\mathbf{X_p}$ *could have been sampled. Therefore, the joint proposal distribution is given by* $Q(\mathbf{x_p}) = \prod_{i=1}^{|\mathbf{X_p}|} Q_{X_{p_i}|\mathbf{x}_{p_{1:i-1}}}(x_{p_i}|\mathbf{x}_{p_{1:i-1}})$.

Hence, computing the joint proposal $Q(\mathbf{x_p})$ becomes easy given a deterministic selection policy $\pi$.

## 3 Collapsed Compilation

Online collapsed importance sampling presents us with a powerful technique for adapting to problems traditional collapsed importance sampling may struggle with. However, it also demands we solve several difficult tasks: one needs a good proposal distribution over any subset of variables, an efficient way of exactly computing an expectation given a sample, and an efficient way of finding the true probability of sampled variables. In this section, we introduce collapsed compilation, which tackles all three of these problems at once using techniques from knowledge compilation.

### 3.1 Knowledge Compilation Background

We begin with a short review of how to perform exact inference on a probabilistic graphical model using knowledge compilation to arithmetic circuits (ACs).

Suppose we have a factor graph (Koller & Friedman, 2009) consisting of three binary variables $A$, $B$ and $C$, and factors $f_1$, $f_2$ as depicted in Figure 3. Each of these factors, as well as their product can be represented as an arithmetic circuit. These circuits have inputs corresponding to variable assignments (e.g., $A$ and $\neg A$) or constants (e.g., 5). Internal nodes are sums or products. We can encode a complete instantiation of the random variables by setting the corresponding variable assignments to 1 and the opposing assignments to 0. Then, the root of the circuit for a factor evaluates to the value of the factor for that instantiation. However, ACs can also represent products of factors. In that case, the AC's root evaluates to a weight that is the product of factor values. Under factor graph semantics, this weight represents the unnormalized probability of a possible world.

The use of ACs for probabilistic inference stems from two important properties. Product nodes are *decomposable*, meaning that their inputs are disjoint, having no variable inputs in common. Sum nodes are *deterministic*, meaning that for any given complete input assignment to the circuit, at most one of the sum's inputs evaluates to a non-zero value. Because of decomposability, we are able to perform marginal inference on ACs: by setting both assignments for the same variable to 1, we effectively marginalize out that variable. For example, by setting all inputs to 1, the arithmetic circuit evaluates to the sum of weights of all worlds, which is the partition function of the graphical model. We refer to Darwiche (2009) for further details on how to reason with arithmetic circuits.

In practice, arithmetic circuits are often compiled from graphical models by encoding graphical model inference into a logical task called weighted model counting, followed by using Boolean circuit compilation techniques on the weighted model counting problem. We refer to Choi et al. (2013) and Chavira & Darwiche (2008) for details. As our Boolean circuit compilation target, we will use the sentential decision diagram (SDD) (Darwiche, 2011). Given any two SDDs representing

factors $f_1, f_2$, we can efficiently compute the SDD representing the factor multiplication of $f_1$ and $f_2$, as well as the result of conditioning the factor graph on any instantiation $x$. We call such operations APPLY, and they are the key to using knowledge compilation for doing online collapsed importance sampling. An example of multiplying two arithmetic circuits is depicted in Figure 3.

As a result of SDDs supporting the APPLY operations, we can directly compile graphical models to circuits in a bottom-up manner. Concretely, we start out by compiling each factor into a corresponding SDD representation using the encoding of Choi et al. (2013). Next, these SDDs are multiplied in order to obtain a representation for the entire model. As shown by Choi et al. (2013), this straightforward approach can be used to achieve state-of-the-art exact inference on probabilistic graphical models.

### 3.2 Algorithm

Now that we have proposed online collapsed importance sampling and given background on knowledge compilation, we are ready to introduce collapsed compilation, an algorithm that uses knowledge compilation to do online collapsed importance sampling.

Collapsed compilation begins by multiplying factors represented as SDDs. When the resulting SDD becomes too large, we invoke online collapsed importance sampling to instantiate one of the variables. On the arithmetic circuit representation, sampling a variable replaces one input by 1 and the other by 0. This conditioning operation allows us to simplify the SDD until it is sufficiently small again. At the end, the sampled variables form $\mathbf{x_p}$, and the variables remaining in the SDD form $\mathbf{X_d}$.

Concretely, collapsed compilation repeatedly performs a few simple steps, following Algorithm 1:

1. Choose an order, and begin multiplying compiled factors into the current SDD until the size limit is reached.

2. Select a variable $X$ using the given policy $\pi$.

3. Sample $X$ according to its marginal probability in the current SDD, corresponding to the partially compiled factor graph conditioned on prior instantiations.

4. Condition the SDD on the sampled value for $X$.

We are taking advantage of knowledge compilation in a few subtle ways. First, to obtain the importance weights, we compute the partition function on the final resulting circuit, which corresponds to the unnormalized probability of all sampled variables, that is, $\hat{P}(\mathbf{x_p})$ in Algorithm 1. Second, Step 3 presents a non-trivial and effective proposal distribution, which due to the properties of SDDs is efficient to compute in the size of the circuit. Third, all APPLY operations on SDDs can be performed tractably (Van den Broeck & Darwiche, 2015), which allows us to multiply factors and condition SDDs on sampled instantiations.

The full technical description and implementation details can be found in Appendix B and C.

## 4   Experimental Evaluation

**Data & Evaluation Criteria**   To empirically investigate collapsed compilation, we evaluate the performance of estimating a single marginal on a series of commonly used graphical models. Each model is followed in parentheses by its number of random variable nodes and factors.

From the 2014 UAI inference competition, we evaluate on linkage(1077,1077), Grids(100,300), DBN(40, 440), and Segmentation(228,845) problem instances. From the 2008 UAI inference competition, we use two semi-deterministic grid instances, 50-20(400, 400) and 75-26(676, 676). Here the first number indicates the percentage of factor entries that are deterministic, and the second indicates the size of the grid. Finally, we generated a randomized frustrated Ising model on a 16x16 grid, frust16(256, 480). Beyond these seven benchmarks, we experimented on ten additional standard benchmarks. Because those were either too easy (showing no difference between collapsed compilation and the baselines), or similar to other benchmarks, we do not report on them here.

For evaluation, we run all sampling-based methods 5 times for 1 hour each. We report the median Hellinger distance across all runs, which for discrete distributions $P$ and $Q$ is given by

$$H(P,Q) = \frac{1}{\sqrt{2}}\sqrt{\sum_{i=1}^{k}(\sqrt{p_i} - \sqrt{q_i})^2}.$$

**Compilation Order**   Once we have compiled an SDD for each factor in the graphical model, collapsed compilation requires us to choose in which order to multiply these SDDs. We look at two orders: BFS and revBFS. The first begins from the marginal query variable, and compiles outwards in a breadth-first order. The second does the same, but in exactly the opposite order arriving at the query variable last.

**Variable Selection Policies**   We evaluate three variable selection policies:

The first policy RBVar explores the idea of picking the variable that least increases the Rao-Blackwell variance of the query (Darwiche, 2009). For a given query $\alpha$, to select our next variable from $\mathbf{X}$, we use $\operatorname{argmin}_{X \in \mathbf{X}} \sum_x P(\alpha|X)^2 P(X)$. This quantity can be computed in time linear in the size of the current SDD.

The next policy we look at is MinEnt, which selects the variable with the smallest entropy. Intuitively, this is selecting the variable for which sampling assumes the least amount of unknown information.

Finally, we examine a graph-based policy FD (FrontierDistance). At any given point in our compilation we have some frontier $\mathcal{F}$, which is the set of variables that have some but not all factors included in the current SDD. Then we select the variable in our current SDD that is, on the graph of our model, closest to the "center" induced by $\mathcal{F}$. That is, we use $\operatorname{argmin}_{X \in \mathbf{X}} \max_{F \in \mathcal{F}} \operatorname{distance}(X, F)$.

In our experiments, policy RBVar is used with the compilation order BFS, while policies MinEnt and FrontierDist are used with order RevBFS.

## 4.1   Understanding Collapsed Compilation

We begin our evaluation with experiments designed to shed some light on different components involved in collapsed compilation. First, we evaluate our choice in proposal distribution by comparison to marginal-based proposals. Then, we examine the effects of setting different size thresholds for compilation on the overall performance, as well as the sample count and quality.

**Evaluating the Proposal Distribution**   Selecting an effective proposal distribution is key to successfully using importance sampling estimation (Tokdar & Kass, 2010). As discussed in Section 3, one requirement of online collapsed importance sampling is that we must provide a proposal distribution over any subset of variables, which in general is challenging.

To evaluate the quality of collapsed compilation's proposal distribution, we compare it to using marginal-based proposals, and highlight the problem with such proposals. First, we compare to a dummy uniform proposal. Second, we compare to a proposal that uses the true marginals for each variable. Experiments on the 50-20 benchmark are shown in Table 1a. Note that these experiments were run for 3 hours rather than 1 hour, so the numbers can not be compared exactly to other tables.

Particularly with policies FrontierDist and MinEnt, the results underline the effectiveness of collapsed compilation's proposal distribution over baselines. This is the effect of conditioning – even sampling from the true posterior marginals does not work very well, due to the missed correlation between variables. Since we are already conditioning for our partial exact inference, collapsed compilation's proposal distribution is providing this improvement for very little added cost.

**Choosing a Size Threshold**   A second requirement for collapsed compilation is to set a size threshold for the circuit being maintained. Setting the threshold to be infinity leaves us with exact inference which is in general intractable, while setting the threshold to zero leaves us with importance sampling using what is likely a poor proposal distribution (since we can only consider one factor at a time). Clearly, the optimal choice finds a trade-off between these two considerations.

Using benchmark 50-20 again, we compare the performance on three different settings for the circuit size threshold: 10,000, 100,000, and 1,000,000. Table 1b shows that generally, 100k gives the best

Table 1: Internal comparisons for collapsed compilation. Values represent Hellinger distances.

(a) Comparison of proposal distributions

| Policy | Dummy | True | SDD |
|---|---|---|---|
| FD | 2.37e−4 | 1.77e−4 | 3.72e−7 |
| MinEnt | 3.29e−4 | 1.31e−3 | 2.10e−8 |
| RBVar | 5.81e−3 | 5.71e−3 | 7.34e−3 |

(b) Comparison of size thresholds

| Policy | 10k | 100k | 1m |
|---|---|---|---|
| FD | 7.33e−5 | 9.77e−6 | 7.53e−6 |
| MinEnt | 1.44e−3 | 1.50e−5 | 8.07e−4 |
| RBVar | 2.96e−2 | 2.66e−2 | 8.81e−3 |

(c) Comparison of size thresholds (50 samples)

| Policy | 10k | 100k | 1m |
|---|---|---|---|
| FD | 1.63e−3 | 5.08e−7 | 1.27e−6 |
| MinEnt | 1.69e−2 | 1.84e−6 | 7.24e−6 |
| RBVar | 1.94e−2 | 1.52e−1 | 3.07e−2 |

(d) Number of samples taken in 1 hour by size

| Size Threshold | 10k | 100k | 1m |
|---|---|---|---|
| Number of Samples | 561.3 | 33.5 | 4.7 |

performance, but the results are often similar. To further investigate this, Table 1c and Table 1d show performance with exactly 50 samples for each size, and number of samples per hour respectively. This is more informative as to why 100k gave the best performance - there is a massive difference in performance for a fixed number of samples between 10k and 100k or 1m. The gap between 100k and 1m is quite small, so as a result the increased number of samples for 100k leads to better performance. Intuitively, this is due to the nature of exact circuit compilation, where at a certain size point of compilation you enter an exponential regime. Ideally, we would like to stop compiling right before we reach that point. Thus, we proceed with 100k as our size-threshold setting for further experiments.

## 4.2   Memory-Constrained Comparison

In this section, we compare collapsed compilation to two related state-of-the-art methods: edge-deletion belief propagation (EDBP) (Choi & Darwiche, 2006), and IJGP-Samplesearch (SS) (Gogate & Dechter, 2011). Generally, for example in past UAI probabilistic inference competitions, comparing methods in this space involves a fixed amount of time and memory being given to each tool. The results are then directly compared to determine the empirically best performing algorithm. While this is certainly a useful metric, it is highly dependent on efficiency of implementation, and moreover does not provide as good of an understanding of the effects of being allowed to do more or less exact inference. To give more informative results, in addition to a time limit, we restrict our comparison at the algorithmic level, by controlling for the level of exact inference being performed.

**Edge-Deletion Belief Propagation**   EDBP performs approximate inference by increasingly running more exact junction tree inference, and approximating the rest via belief propagation (Choi & Darwiche, 2006; Choi et al., 2005). To constrain EDBP, we limit the corresponding circuit size for the junction tree used. In our experiments we set these limits at 100,000 and 1,000,000.

**IJGP-Samplesearch**   IJGP-Samplesearch (SS) is an importance sampler augmented with constraint satisfaction search (Gogate & Dechter, 2011, 2007). It uses iterative join-graph propagation (Dechter et al., 2002) together with $w$-cutset sampling (Bidyuk & Dechter, 2007) to form a proposal, and then uses search to ensure that no samples are rejected. To constrain SS, we limit treewidth $w$ at either 15, 12, or 10. For reference, a circuit of size 100,000 corresponds to a treewidth between 10 and 12.

Appendix D describes both baselines as well as the experimental setup in further detail.

### 4.2.1   Discussion

Table 2 shows the experimental results for this setting. Overall, we have found that when restricting all methods to only do a fixed amount of exact inference, collapsed compilation has similar performance to both Samplesearch and EDBP. Furthermore, given a good choice of variable selection policy, it can often perform better. In particular, we highlight DBN, where we see that collapsed compilation with the RBVar or MinEnt policies is the only method that manages to achieve reasonable approximate inference. This follows the intuition discussed in Section 2.2: a good choice of a few variables in a densely connected model can lead to relatively easy exact inference for a large chunk of the model.

Table 2: Hellinger distances across methods with internal treewidth and size bounds

| Method | 50-20 | 75-26 | DBN | Grids | Segment | linkage | frust |
|---|---|---|---|---|---|---|---|
| EDBP-100k | 2.19e−3 | 3.17e−5 | 6.39e−1 | 1.24e−3 | 1.63e−6 | 6.54e−8 | 4.73e−3 |
| EDBP-1m | 7.40e−7 | 2.21e−4 | 6.39e−1 | 1.98e−7 | 1.93e−7 | 5.98e−8 | 4.73e−3 |
| SS-10 | 2.51e−2 | 2.22e−3 | 6.37e−1 | 3.10e−1 | 3.11e−7 | 4.93e−2 | 1.05e−2 |
| SS-12 | 6.96e−3 | 1.02e−3 | 6.27e−1 | 2.48e−1 | 3.11e−7 | 1.10e−3 | 5.27e−4 |
| SS-15 | 9.09e−6 | 1.09e−4 | (Exact) | 8.74e−4 | 3.11e−7 | 4.06e−6 | 6.23e−3 |
| FD | 9.77e−6 | 1.87e−3 | 1.24e−1 | 1.98e−4 | 6.00e−8 | 5.99e−6 | 5.96e−6 |
| MinEnt | 1.50e−5 | 3.29e−2 | 1.83e−2 | 3.61e−3 | 3.40e−7 | 6.16e−5 | 3.10e−2 |
| RBVar | 2.66e−2 | 4.39e−1 | 6.27e−3 | 1.20e−1 | 3.01e−7 | 2.02e−2 | 2.30e−3 |

Another factor differentiating collapsed compilation from both EDBP and Samplesearch is the lack of reliance on some type of belief propagation algorithm. Loopy belief propagation is a cornerstone of approximate inference in graphical models, but it is known to have problems converging to a good approximation on certain classes of models (Murphy et al., 1999). The problem instance frust16 is one such example – it is an Ising model with spins set up such that potentials can form loops, and the performance of both EDBP and Samplesearch highlights these issues.

## 4.3 Probabilistic Program Inference

As an additional point of comparison, we introduce a new type of benchmark. We use the probabilistic logic programming language ProbLog (De Raedt & Kimmig, 2015) to model a graph with probabilistic edges, and then query for the probability of two nodes being connected. This problem presents a unique challenge, as every non-unary factor is deterministic.

| Method | Prob12 |
|---|---|
| EDBP-1m | 3.18e−1 |
| SS-15 | 3.87e−3 |
| FD | 1.50e−3 |

Table 3: Hellinger distances for ProbLog benchmark

Table 3 shows the results for this benchmark, with the underlying graph being a 12x12 grid. We see that EDBP struggles here due to the large number of deterministic factors, which stop belief propagation from converging in the allowed number of iterations. Samplesearch and collapsed compilation show similarly decent results, but interestingly they are not happening for the same reason. To contextualize this discussion, consider the stability of each method. Collapsed compilation draws far fewer samples than SS – some of this is made up for by how powerful collapsing is as a variance reduction technique, but it is indeed less stable than SS. For this particular instance, we found that while different runs for collapsed compilation tended to give different marginals fairly near the true value, SS consistently gave the same incorrect marginal. This suggests that if we ran each algorithm until convergence, collapsed compilation would tend toward the correct solution, while SS would not, and appears to have a bias on this benchmark.

## 5 Related Work and Conclusions

We have presented online collapsed importance sampling, an asymptotically unbiased estimator that allows for doing collapsed importance sampling without choosing which variables to collapse a priori. Using techniques from knowledge compilation, we developed collapsed compilation, an implementation of online collapsed importance sampling that draws its proposal distribution from partial compilations of the distribution, and naturally exploits structure in the distribution.

In related work, Lowd & Domingos (2010) study arithmetic circuits as a variational approximation of graphical models. Approximate compilation has been used for inference in probabilistic (logic) programs (Vlasselaer et al., 2015). Other approximate inference algorithms that exploit local structure include samplesearch and the family of universal hashing algorithms (Ermon et al., 2013; Chakraborty et al., 2014). Finally, collapsed compilation can be viewed as an approximate knowledge compilation method: each drawn sample presents a partial knowledge base along with the corresponding correction weight. This means that it can be used to approximate any query which can be performed efficiently on an SDD – for example the most probable explanation (MPE) query (Chan & Darwiche, 2006; Choi & Darwiche, 2017). We leave this as an interesting direction for future work.

**Acknowledgements**

We thank Jonas Vlasselaer and Wannes Meert for initial discussions. Additionally, we thank Arthur Choi, Yujia Shen, Steven Holtzen, and YooJung Choi for helpful feedback. This work is partially supported by a gift from Intel, NSF grants #IIS-1657613, #IIS-1633857, #CCF-1837129, and DARPA XAI grant #N66001-17-2-4032.

## Footnotes

[1]The code is available at `https://github.com/UCLA-StarAI/Collapsed-Compilation`. It uses the SDD library for knowledge compilation (Darwiche, 2011) and the Scala interface by Bekker et al. (2015).

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
