[Supplementary Material · nips_2018_supp.pdf]

# A  Proof of Theorems

We begin with the formal definition of our augmented factor graph.

**Definition 4.** *Suppose we have a discrete distribution represented by a factor graph $F$, with variables $\mathbf{X}$. Then we define the corresponding augmented factor graph $F_A$ as follows:*

- *For each $X_i$, we introduce variables $X_i^*$ and $S_i$*

- *For each $S_i$, add a unary factor such that $S_i$ is uniform*

- *For each $X_i^*$, add a factor such that*

$$P(X_i^* | S_i, X_i) = \begin{cases} 0.5, & \text{if } S_i = 0 \\ 1, & \text{if } S_i = 1, X_i = X_i^* \\ 0, & \text{otherwise} \end{cases} \tag{3}$$

## A.1  Proof of Theorem 1

*Proof.* Consider some variable $X_i$ together with its corresponding augmented variables $X_i^*$, $S_i$. Then examining the resulting graphical model, we see that $X*_i$ and $S_i$ are only connected to the rest of the model via $X_i$. Due to the conditional independence properties, this means that we can first sum out $S_i$ from $P(X_i^* | S_i, X_i)$, and then simply sum $X_i^*$. Thus, the result of any query over $\mathbf{X}$ on $F_A$ will be equivalent to the result on $F$, as we can simply sum out all additional variables, and end up at the same model. $\square$

## A.2  Proof of Theorem 2

NB: This section only provides the proof for the case of binary variables. When compiling graphical models with multivalued variables, it is done by assuming an indicator variable for each possible state of each variable, and then adding logical constrains such that exactly one of these indicators is true at any time. Thus, proving the binary case is sufficient.

*Proof.* Our goal is to estimate

$$\mathbb{E}_{P(\xi)}[f(\xi)] = \sum_{X, X^*, S} P(X, X^*, S) f(X)$$

$$= \sum_{X, X^*, S} P(X) \prod_i P(S_i) P(X_i^* | X_i, S_i) f(X)$$

We begin by introducing our proposal distribution $Q(X^*, S)$, and supposing we have samples $\{X^*[m], S[m]\}_{m=1}^M$ drawn from our proposal distribution $Q$. Now, substituting this in and continuing our previous equations gives us:

$$\sum_{X^*[m], S[m]} \frac{1}{Q(X^*[m], S[m])} \sum_X P(X) \prod_i P(S_i[m]) P(X_i^*[m]) f(X).$$

We can split our sum over $X$ up into indices where $S_i[m] = 0$, and ones where $S_i[m] = 1$. For the sake of notation, we will refer to these sets of indices as $I^C$ and $I$ respectively:

$$\sum_{m=1}^M \frac{1}{Q(X^*[m], S[m])} \sum_{X_I} \prod_{i \in I} (P(S_i[m]) P(X_i^*[m] | X_i, S_i[m]))$$

$$\sum_{X_{I^C}} P(X) \prod_{i \in I^C} P(S_i[m]) P(X_i^*[m] | X_i, S_i[m]) f(X)$$

$$= \sum_{m=1}^M \frac{1}{Q(X^*[m], S[m])} \sum_{X_I} \prod_{i \in I} (P(S_i[m]) P(X_i^*[m] | X_i, S_i[m]))$$

$$\sum_{X_{I^C}} P(X_I)P(X_{I^C}|X_I) \prod_{i \in I^C}(0.5)P(X_i^*[m]|X_i, S_i[m])f(X)$$

$$= \sum_{m=1}^{M} \frac{1}{Q(X^*[m], S[m])} \sum_{X_I} P(X_I) \prod_{i \in I}(P(S_i[m])P(X_i^*[m]|X_i, S_i[m]))$$

$$\sum_{X_{I^C}} P(X_{I^C}|X_I)(0.5 \cdot 0.5)^{|I^C|} f(X)$$

$$= \sum_{m=1}^{M} \frac{(0.25)^{|I^C|}}{Q(X^*[m], S[m])} \sum_{X_I} P(X_I) \prod_{i \in I}(0.5)P(X_i^*[m]|X_i, S_i[m]) \sum_{X_{I^C}} P(X_{I^C}|X_I)f(X)$$

$$= \sum_{m=1}^{M} \frac{(0.25)^{|I^C|}}{Q(X^*[m], S[m])}(0.5)^{|I|} \sum_{X_I} P(X_I) \prod_{i \in I} P(X_i^*[m]|X_i, S_i[m]) \mathbb{E}_{P(X_{I^C}|X_I)}[f(X)]$$

Now, observe that the term $\prod_{i \in I} P(X_i^*[m]|X_i, S_i[m]) = 1$ if and only if $\forall i$ s.t. $S_i[m] = 1$, we have that $X_i = X_i^*[m]$. Since there is only one setting of these indices that satisfies this (that is, letting $X_i = X_i^*[m]$ everywhere that $S_i[m] = 1$), this allows us to remove this sum, and obtain:

$$\sum_{m=1}^{M} \frac{(0.25)^{I^C}(0.5)^{I} P_{X_I}(X_I^*[m]) \mathbb{E}_{P(X_{I^C}|X_I)}[f(X)]}{Q(X^*[m], S[m])},$$

which is precisely what we wanted – this is the equation we would expect to use for our online collapsed importance sampler (once we adjust our proposal distribution for variables that are not actually sampled to correct for the 0.5s that are left). $\qquad \square$

### A.3  Proof of Theorem 3

*Proof.* We will proceed by contradiction. Suppose we have two paths through our variables, $X_{i_1}^*, X_{i_2}^*, \ldots, X_{i_n}^*$ and $X_{j_1}^*, X_{j_2}^*, \ldots, X_{j_n}^*$ which can produce the same assignment to all variables. Now, there are two key facts we will observe and make use of:

1. The starting point for any path is fixed, that is $X_{i_1}^* = X_{j_1}^*$. Our heuristics are deterministic, and the memory limit remains constant, so the first variable to be sampled is always the same.

2. Once an assignment is made to the current variable being sampled, the decision of which variable to sample is deterministic – again, because our heuristics must be deterministic. To put it another way, if $x_{i_k}^* = x_{j_k}^*$, then $X_{i_k+1}^* = X_{j_k+1}^*$.

Now, since path $i$ and path $j$ are different, there must be some first element $k$ where $X_{i_k}^* \neq X_{j_k}^*$. By Fact 1, $k > 1$. Also, observe that since $k$ is the first such element, $X_{i_{k-1}}^* = X_{j_{k-1}}^*$. But since our two paths must give the same assignment to all variables, this means also that $x_{i_{k-1}}^* = x_{j_{k-1}}^*$, which means that $X_{i_k}^* = X_{j_k}^*$ by Fact 2. This is a contradiction.

$\qquad \square$

## B  Collapsed Compilation: Algorithm Outline

Algorithm 2 describes Collapsed Compilation in more detail. Note that to avoid confusion, we hereafter refer to the process of sampling $x \sim Q(X)$ as conditioning (since all future variables sampled are conditioned on $x$), and a single full run as a sample. Conditioning an SDD on instantiation $x$ is denoted $SDD|x$.

There are a few important things to take away here. First, if we at any point interrupt bottom-up compilation, what we will get is a complete compilation of some subset of the model. This means that on Line 8, the proposal distribution $P_{SDD}$ we are drawing from is the true distribution for $X_j$ on some subset of graphical model $M$, conditioned on all previous variables.

---

**Algorithm 2:** A single sample of Collapsed Compilation

---

**Input :** $\pi$: A variable selection policy computed using an SDD,
$\quad\quad\quad$ $M$: A probabilistic graphical model,
$\quad\quad\quad$ $f$: Function to estimate expectation for

**Result:** A single sample $\left(\mathbf{x_p^m}[m], \ \mathbb{E}_{P(\mathbf{X_d^m}|\mathbf{x_p^m}[m])}[f(\mathbf{x_p^m}[m], \mathbf{X_d^m})], \ w[m]\right)$

---

**1** $SDD \leftarrow True$
**2** $\mathbf{x_p} \leftarrow \{\}$
**3** $q \leftarrow 1$
**4** **while** *$M$ is not compiled* **do**
**5** $\quad$ $SDD \leftarrow bottomUpCompileStep(SDD, M)$
**6** $\quad$ **while** *SDD is too large* **do**
**7** $\quad\quad$ $X \leftarrow \pi(SDD, \mathbf{x_p})$
**8** $\quad\quad$ $x \sim P_{SDD}(X)$
**9** $\quad\quad$ $q \leftarrow q \cdot P_{SDD}(x)$
**10** $\quad\quad$ $\mathbf{x_p} \leftarrow \mathbf{x_p} \cup \{X = x\}$
**11** $\quad\quad$ $SDD \leftarrow SDD|x$

**12** **return** $\left(\mathbf{x_p}, \frac{WMC_f(SDD)}{WMC(SDD)}, \frac{WMC(SDD)}{q}\right)$

---

Second, there are a few calls to a weighted model count function $WMC$ on Line 12. Recall that for an SDD representing a probability distribution $P(\mathbf{X})$, the weighted model count subject to a function $f$ computes $\sum_{\mathbf{x}} P(\mathbf{x})f(\mathbf{x})$. Also, observe that the SDD we are left with when we finish compiling is representing the joint distribution $P(\mathbf{X_d}, \mathbf{X_p} = \mathbf{x_p})$. Thus, observing that $P(\mathbf{X_d}, \mathbf{X_p} = \mathbf{x_p}) = \hat{P}(\mathbf{X_d}|\mathbf{X_p} = \mathbf{x_p})$ we see that $WMC_f$ – the weighted model count subject to $f$ – is actually $\sum_{\mathbf{x_d}} \hat{P}(\mathbf{X_d} = \mathbf{x_d}|\mathbf{X_p} = \mathbf{x_p})f(\mathbf{x_d}, \mathbf{x_p})$. But setting $f = 1$ allows us to compute the normalization constant, meaning that $\frac{WMC_f(SDD)}{WMC(SDD)} = \mathbb{E}_{P(\mathbf{X_d}|\mathbf{X_p}=\mathbf{x_p})}[f(\mathbf{x_p}, \mathbf{X_d})]$.

## C Collapsed Compilation: Algorithmic Details

There are many moving parts in this method, so in this section we will examine each in isolation.

### C.1 Choices of Strategies

**Compilation Order** The main paper describes the BFS and revBFS compilation orders.

**Proposal Distribution** Given that we have decided to condition on a variable $X_j$, we decide its proposal distribution by computing the marginal probability of $X_j$ in our currently compiled SDD. This can be done in time linear in the size of the circuit by computing the partial derivative of the weighted model count in the current circuit w.r.t. $X_j$ (Darwiche, 2003).

**Variable Selection** The manner in which we select the next variable to condition on – that is, our choice of $\pi$ in Algorithm 2 – has a large effect on both the tractability and efficiency of our sampler. The main paper defines three policies, the first of which depends specifically on the marginal being queried for, while the other two do not. The policies all satisfy Definition 3: they are deterministic.

### C.2 Determinism

A desirable property for samplers – particularly when there are a large number of deterministic relationships present in the model – is to be *rejection-free*. It is clear that in the absence of deterministic factors (that is, no 0 entries in any factor), collapsed compilation will never reject samples. Here, we describe how this guarantee can be maintained in the presence of 0-valued factor entries.

**Extracting a Logical Base** Suppose we are given a factor over some variables $X_1, X_2..., X_n$. Then a weight of 0 given to an assignment $x_1, x_2.., x_n$ indicates a logical statement. Specifically, we can

say for certain over the entire model that $\neg(x_1 \wedge x_2, ..., \wedge x_n)$. As a preprocessing step, we find all such factor entries in the graph, convert them each into the corresponding logical statement, and then take the conjunction of all of these sentences. This forms the logical base for our model.

**An Oracle for Determinism**    Once we have obtained this base for our model, we can precompile a circuit representing it. This allows us to make queries asking whether there is a non-zero probability that $X = x$, given all previous assignments $\mathbf{x_p}$ (Darwiche, 1999). Thus, when we sample from the marginal probability of $X$ from our current SDD (our proposal distribution), we first renormalize this marginal to only include assignments which have a non-zero probability according to our determinism oracle. Of course, this is not always possible due to size constraints in the case where there is an enormous amount of determinism. For these cases we just run collapsed compilation as is – depending on the compilation order it will still tend to reject few samples.

**Literal Entailment**    As a simple optimization, we can recognize any variables whose values are already deterministically chosen based on previously conditioned variables, and assign them as such in our SDD. Given a determinism oracle, deciding this can be done for all variables in the model in time linear in the size of the oracle (Darwiche, 2001).

# D    Experimental Details

## D.1    Edge-Deletion Belief Propagation

Edge-deletion belief propagation (EDBP) is a method for doing approximate graphical model inference by using a combination of exact inference and belief propagation (Choi & Darwiche, 2006) (Choi et al., 2005). EDBP iteratively computes more and more of the model exactly using junction tree, at each step performing belief propagation to approximate the rest of the model. It can be viewed as the belief propagation analog of collapsed compilation, which makes it an interesting target for comparison. A major conceptual difference between the two is that while collapsed compilation is asymptotically unbiased and thus will given an accurate result given enough time, EDBP will tend to finish more quickly but has no way to improve once converged.

To capture a more direct comparison of the amount of exact inference being performed, we compare collapsed compilation to EDBP with the size of the junction tree used directly being limited, rather than the computer memory usage. In particular, we limit the size of the circuit corresponding to the junction tree to be similar to the sizes used for collapsed compilation. To this end, we use 100,000 and 1,000,000 as size limits for the junction tree, and otherwise run the algorithm as usual. Table 2 shows the results across all benchmarks. Keeping in mind that all policies for collapsed compilation use 100,00 as their size limit, collapsed compilation is comparable to EDBP. Both perform very well in linkage and Segment, and while collapsed compilation performs better on 50-20, EDBP does better on 75-26.

## D.2    SampleSearch

IJGP-Samplesearch is an importance sampler augmented with constraint satisfaction search (Gogate & Dechter, 2011) (Gogate & Dechter, 2007). It uses iterative join graph propagation (Dechter et al., 2002) together with $w$-cutset sampling (Bidyuk & Dechter, 2007) to form a proposal, and then uses search to ensure that no samples are rejected.

Once again, we would like to control for the amount of exact inference being done directly at the algorithmic level, rather than via computer memory. For samplesearch, we do this by limiting $w$, which is the largest treewidth that is allowed when using collapsing to reduce variance. We run samplesearch with three different settings, limiting $w$ to 15, 12, and 10 respectively. Table 2 shows the results of running our standard set of benchmarks with all of these settings. As a reference point, empirically a circuit size limit of 100,00 generally corresponds to a treewidth somewhere between 10 and 12. The results are generally similar to constraining the memory of EDBP, but with more constrained versions of samplesearch suffering more. For example, although linkage appears to be an easy instance in general, without a large enough $w$-cutset, samplesearch struggles compared to other methods.