[Reviews · NeurIPS 2018]

Reviewer 1



The paper proposes an approach for approximate inference in discrete graphical models that generalizes collapsed importance sampling. The apprach is online, in the sense that the choice of variables to sample depends on the values already sampled, instead of being fixed. Moreover, the exact inference portion of the method is performed by knwoledge compilation so the approach is called collapsed compilation and allows approximate knowledge compilation. The method begins by multiplying factors obtaining a Sentential Decision Diagram. When the SDD becomes too large, a variable is deterministically chosen, a value for it is sampled from a proposal distribution that is the marginal of the variable in the current SDD and the SDD is conditioned on the sampled value. This is repeated until all factors have been taken into account. Collapsed compilation is extensively experimentally compared to Edge-Deletion Belief Propagation and IJGP-Samplesearch. The results show that it often performs better on various domains from UAI inference competition and probabilistic logic programming. The method is put correctly in the context of related work such as Lowd & Domingos (2010) and (Ermon et al., 2013; Chakraborty 325 et al., 2014). The paper provides a very interesting new approach for mixing approximate and exact inference. The paper is overall clear but some information on the heuristics that is now in the supplementary material would be more useful in the main paper. In particular, Sections C.1-3 on compilation order, proposal distribution and variable selection provide important information on the method. You can make room for it in the paper by simplifying Section 2: Theorem 2 shows that the variables that are sampled can differ in different samples so that eq 2 in unbiased. You can simply state the theorem this way without the need to introduce the augmented factor graph whose usefulness is limited if the proof of the theorem is not presented together with it. Moreover, since you consider only deterministic strategies, the presentation may be simplified by discussing only those, without introducing stochastic variables selection policies, that can be put in the supplementary material. The paper seems technically sound, the proofs in the appendix seem correct with a few minor glitches: 1) the proof of the theorems are limited to binary variables 2) in Algorithm 2, I believe the third return value should be WMC(SDD)/q, not q/WMC(SDD) 3) the first equation in A.2 is missing f(X) 4) def 2: given M samples 5) page 12: X_I is missing [m] in the equations The paper provides an original approach for combining exact and approximate inference by sampling, integrating collapsed sampling with exact inference by knowledge compilation, using a clever proposal distribution that is close to the correct distribution yet easy to compute. The significance of the method is shown by the experiments, that demonstrate the effectiveness of the approach in comparison with the state of the art. I have a question that I would like the authors to answer in the response: would it be useful to resuse the compiled factors across samples? It seems to me that it could significantly improve the efficiency. After reading the other reviews and the authors' feedback, I think all questions put forward by the reviewers were adequately answered, so the paper for me should be accepted.

Reviewer 2



This paper proposes an approximate inference algorithm that combines SDDs with importance sampling. Previously, SDDs have been used for exact inference, by compiling probabilistic graphical models into a tractable SDD representation for answering queries. However, this compilation may lead to an exponentially large representation. Instead, the authors of this paper propose to interleave the compilation process with sampling assignments for individual variables in order to reduce the SDD and keep its size small. The partially compiled SDD itself serves as a proposal distribution for sampling. The algorithm is similar to other importance sampling algorithms, but I thought the way that it used SDDs and sampled variables dynamically was clever. The experiments comparing different variable selection strategies and baseline proposal distributions were useful. It makes sense that the frontier method would work well, since variables that participate in many factors could have their probabilities greatly changed by each factor. In fact, you could design really bad cases where factors contradict the included ones, leading to a proposal distribution that's much worse than uniform. I'd be interested to see additional discussion of limitations and when these methods will work particularly well or poorly compared to other algorithms. I do not follow the literature on approximate inference closely enough to know if the baselines used are still state-of-the-art and represent the best comparison. SampleSearch and EDBP have been around for a while... is there anything more recent that's worthy of discussion or empirical comparison? That said, both algorithms have performed well on these benchmarks in the past, and the benchmarks are from inference competitions designed specifically to test approximate inference algorithms. In my review, I will assume that the empirical comparison is as strong as reported. Additional discussion of related work (even if it's not compared empirically) would be welcome, as would more explanation of why these inference tasks are meaningful on these datasets and how to interpret the metrics. For example, collapsed inference means that probabilities of very unlikely events can still be estimated. Are there appropriate experiments that could test predictions of rare events? One suggested piece of related work: Gogate and Domingos, "Approximation by Quantization," UAI 2011 This work performs exact inference on a model where the parameters have been approximated to make reasoning tractable. I suspect this paper is less general-purpose and less accurate than what is proposed here, but there are some common themes of connecting exact and approximate inference and using representations that support rich structure (algebraic decision diagrams, in this case, instead of SDDs). (This is not a serious problem -- just a suggestion for the authors which they are free to disregard.) Overall summary: Clever combination of different inference methods with good empirical results. If a longer version of this paper is ever published, it would help to have even more analysis and context. After the author response: I've read the author response and the other reviews, and I remain enthusiastic about this paper. The comments by Reviewer 1 and the author response increase my confidence that this paper is sound and state-of-the-art. The comments by Reviewer 3 do not convince me that anything is lacking or inadequate about this paper. I have changed my rating from 7 to 8.

Reviewer 3



The authors propose an approximate inference algorithm named collapsed compilation based on collapsed importance sampling and knowledge compilation (which is responsible for the exact inference part). It first compiles factors to SDDs based on a specific order to some size limit, and then runs online importance sampling with samples from a proposal distribution aware of the exactness based on the current partially compiled graph. The method is kind of complicated to use in practice since it depends on many parameters (e.g., circuit size, variable selection policy). Those instances reported in the experiment are quite diverse, and the probabilistic program inference benchmark is interesting. The writing is generally good. To the authors: 1) “ we decide its proposal distribution by computing the marginal probability of Xj in our currently compiled SDD” (line 461), so you do not need any external proposal distribution, right? 2) Table 2 provides results on individual instances only, which is not that convincing. Do you have aggregated results for multiple instances of each benchmark? 3) From Table 1(b), we can observe that the “100k” entry for “FD” is better than the “1m” entry, why?